# STAG1 vulnerabilities for exploiting cohesin synthetic lethality in STAG2-deficient cancers

Petra van der Lelij[1],*, Joseph A Newman[2],*, Simone Lieb[3],*, Julian Jude[1],*, Vittorio Katis[2], Thomas Hoffmann[1], Matthias Hinterndorfer[1], Gerd Bader[3], Norbert Kraut[3], Mark A Pearson[3], Jan-Michael Peters[1,4], Johannes Zuber[1,4], Opher Gileadi[2], Mark Petronczki[3]

The cohesin subunit *STAG2* has emerged as a recurrently inactivated tumor suppressor in human cancers. Using candidate approaches, recent studies have revealed a synthetic lethal interaction between *STAG2* and its paralog *STAG1*. To systematically probe genetic vulnerabilities in the absence of STAG2, we have performed genome-wide CRISPR screens in isogenic cell lines and identified STAG1 as the most prominent and selective dependency of STAG2-deficient cells. Using an inducible degron system, we show that chemical genetic degradation of STAG1 protein results in the loss of sister chromatid cohesion and rapid cell death in STAG2-deficient cells, while sparing *STAG2*–wild-type cells. Biochemical assays and X-ray crystallography identify STAG1 regions that interact with the RAD21 subunit of the cohesin complex. STAG1 mutations that abrogate this interaction selectively compromise the viability of *STAG2*-deficient cells. Our work highlights the degradation of STAG1 and inhibition of its interaction with RAD21 as promising therapeutic strategies. These findings lay the groundwork for the development of STAG1-directed small molecules to exploit synthetic lethality in *STAG2*-mutated tumors.

## Introduction

Pan-cancer genome studies have identified recurrent mutations in the cohesin complex and its regulators in a variety of human cancers (Lawrence et al, 2014; Leiserson et al, 2015). Cohesin is a multi-subunit protein complex that is essential for sister chromatid cohesion from DNA replication to mitosis. This function of cohesin is crucial for chromosome segregation and the generation of viable daughter cells during cell division (Guacci et al, 1997; Michaelis et al, 1997). In addition, cohesin has important functions in genome organization, gene regulation, and DNA damage repair (De Koninck

& Losada, 2016; Nishiyama, 2019). The complex consists of a tripartite ring comprising SMC1, SMC3, and RAD21 (also called SCC1 or Mcd1), which associates with a conserved peripheral fourth subunit that, in human somatic cells, is represented by two paralogs of the Scc3/STAG protein family, STAG1 or STAG2 (Losada et al, 2000; Sumara et al, 2000; Roig et al, 2014). STAG1/2 HEAT repeat proteins bind to the RAD21 subunit of the core cohesin ring (Toth et al, 1999; Haering et al, 2002; Hara et al, 2014) and are required for the dynamic association of the complex with chromatin (Hu et al, 2011; Murayama & Uhlmann, 2014; Roig et al, 2014). Cohesin complexes containing STAG1 and STAG2 are particularly important for sister chromatid cohesion at telomeres and centromeres, respectively, whereas both complexes appear to contribute to cohesion along chromosome arms (Canudas & Smith, 2009; Remeseiro et al, 2012). STAG1 and STAG2, therefore, function redundantly in somatic mammalian cells to mediate sister chromatid cohesion, an event essential for cell viability during proliferation.

*STAG2*, which is encoded on the X chromosome, represents the most frequently mutated cohesin subunit in human cancers with predominantly deleterious alterations detected in 15–20% of bladder cancer and Ewing sarcoma tumors and in ~6% of acute myeloid leukemia and myelodysplastic syndrome cases (Solomon et al, 2011; Hill et al, 2016). Therefore, *STAG2* appears to function as a tumor suppressor gene in the affected tissues. How *STAG2* loss-of-function mutations are driving tumorigenesis is poorly understood. The pathological mechanism is thought to be unrelated to defects in sister chromatid cohesion and resulting aneuploidy, as many *STAG2*-mutated tumors have normal karyotypes (De Koninck & Losada, 2016). Instead, cohesin mutations may promote tumor formation by interfering with cohesin's role in gene regulation and genome organization (Thota et al, 2014; Mazumdar et al, 2015; Mullenders et al, 2015; Viny et al, 2015). Irrespective of its role in driving tumorigenesis, the abundance of *STAG2* alterations in cancer has moved the cohesin subunit into the focus of research for new therapeutic concepts in oncology.

[1]Research Institute of Molecular Pathology (IMP), Vienna BioCenter (VBC), Vienna, Austria   [2]Structural Genomics Consortium, University of Oxford, Oxford, UK   [3]Boehringer Ingelheim Regional Center Vienna (RCV) GmbH & Co KG, Vienna, Austria   [4]Medical University of Vienna, VBC, Vienna, Austria

Correspondence: johannes.zuber@imp.ac.at; opher.gileadi@sgc.ox.ac.uk; mark_paul.petronczki@boehringer-ingelheim.com
*Petra van der Lelij, Joseph A Newman, Simone Lieb, and Julian Jude contributed equally to this work

In recent studies, a synthetic lethal interaction between *STAG2* and its paralog *STAG1* has been identified through candidate approaches (Benedetti et al, 2017; van der Lelij et al, 2017; Liu et al, 2018). This genetic interaction is mechanistically explained by the redundancy between STAG1 and STAG2 in mediating sister chromatid cohesion. Losing either paralog is compatible with cohesion, successful chromosome segregation, and cell viability, whereas the concomitant inactivation of both leads to complete loss of sister chromatid cohesion that is detrimental to cell proliferation and survival.

Although a recent study identified DNA repair factors as candidate vulnerabilities in STAG2-deficient cells (Mondal et al, 2019), it is apparent that a comprehensive search for selective dependencies is required to provide fundamental information for developing the most promising therapeutics to treat *STAG2*-mutated tumors. In this study, we used an optimized CRISPR/Cas9–based screening approach and identified STAG1 as the only dependency across the human genome that is entirely selective to *STAG2*-deficient cells. As STAG1 has no known enzymatic activity that could be inhibited, we explored alternative ways to target STAG1 and showed that chemical genetic degradation of STAG1 or inhibiting its interaction with RAD21 are deleterious in *STAG2*-deficient cells, whereas sparing their wild-type counterparts. Our results demonstrate the rationale and lay the groundwork for targeting STAG1 to selectively attack *STAG2*-mutated tumors.

# Results

## STAG1 is the most selective synthetic-lethal dependency of *STAG2*-mutant cells genome wide

CRISPR/Cas9–based dropout screens have emerged as a powerful approach to annotate comprehensively genes required for proliferation and survival (Wang et al, 2015; Meyers et al, 2017). We used this genome-wide approach to expand on candidate-based studies of dependencies of *STAG2*-mutated cells. Considering the technical challenges leading to low signal-to-noise ratios in CRISPR/Cas9–based negative-selection screens (Hinterndorfer & Zuber, 2019), we took several measures to enhance the coverage and dynamic range of our assay. First, we chose the near-haploid KBM-7 human leukemia line to serve as our primary screening model, as functional ablation of the vast majority of genes requires only one CRISPR/Cas9–induced loss-of-function mutation (Burckstummer et al, 2013). Second, we pre-engineered single-cell–derived clones harboring a tetracycline (Tet)-inducible Cas9 expression cassette and selected a clone displaying high CRISPR/Cas9–editing efficacy for the generation of isogenic STAG2-knockout cells and subsequent screens to ensure homogenous Cas9 expression and function (Figs 1A and S1A). Finally, we took advantage of a second-generation genome-wide single gRNA (sgRNA) library (Michlits et al, 2020) and performed screens with high representation (>1,000× infected cells/sgRNA at all steps) in three biological replicates in both *STAG2*–wild-type and *STAG2*-knockout KBM-7 cells (Table S1) (Doench et al, 2014; Wang et al, 2014).

Collectively, these measures yielded high sgRNA dropout effect sizes, facilitating the identification of genes required for proliferation and survival with a high dynamic range (Fig 1B). Negatively

selected genes included a previously defined set of generally essential genes (Wang et al, 2019) that displayed an average depletion of ~24-fold, which exceeds effect sizes observed in previous CRISPR-based negative-selection screens (Wang et al, 2015; Meyers et al, 2017). Dropout effects in isogenic *STAG2*–wild-type and *STAG2*-deficient KBM-7 cells were highly correlated (R = 0.93). The only gene that showed no negative selection in wild-type cells but strong depletion in *STAG2*-mutant cells was the *STAG2* paralog *STAG1* (Fig 1B), which also emerged as the strongest and only significant hit in a statistical analysis of dropout effects in both cell lines (Fig 1C and Table S2). Indeed, in contrast to PLK1, a pan-essential gene, knockout of *STAG1* using three independent sgRNAs was deleterious in *STAG2*-mutant KBM-7 cells used in the screen and an independent *STAG2*-deficient clone, while being neutral in isogenic wild-type cells (Fig 1D). In validation studies using multiple sgRNAs, none of the additional candidates that displayed differential depletion in the pooled primary screen met the strength and selectivity of the synthetic lethality observed between *STAG1* and *STAG2* (Fig S1B). Together with our screening data, these results suggest that *STAG1* is not only a very prominent but also the only hard-wired synthetic-lethal interaction with *STAG2*.

These findings emphasize that targeting STAG1 represents a clear therapeutic opportunity in the treatment of a *STAG2*-mutant cancers in wide range of malignancies. Amongst solid tumors, *STAG2* mutations show the highest prevalence in bladder cancer (Hill et al, 2016). We transplanted *STAG2* mutated (p.K983*) UM-UC-3 bladder cancer cells, which were transduced with in vitro validated shRNAs targeting STAG1, into immunocompromised mice to evaluate the effects of partial STAG1 suppression in vivo (Fig S2A–D). After tumor establishment, shRNA and GFP co-expression was induced by administration of doxycycline. In contrast to tumors expressing a neutral control shRNA, two independent shRNAs targeting STAG1 strongly suppressed tumor growth in vivo (Fig S2E). Importantly, for both STAG1 shRNAs, remaining tumors were mainly composed of GFP-negative cells (Fig S2F), indicating that STAG1 shRNA-expressing cells were strongly selected against. Together with a recent report (Liu et al, 2018), these results demonstrate that partial suppression of STAG1 triggers strong and selective antiproliferative effects in *STAG2*-mutant cancer models in vitro and in vivo and further reinforces the hypothesis that STAG1 should be pursued as a promising concept for therapy development.

## Degradation of STAG1 as a therapeutic strategy

The STAG1 protein is composed of HEAT repeats, a tandem repeat structural motif composed of two *α* helices linked by a short loop. STAG1 has no known enzymatic activity that could be inhibited and there is no precedence for the successful pharmaceutical targeting of HEAT repeats. Advances in small molecule research have led to the discovery of bifunctional compounds capable of pharmacologically inducing target protein degradation, thereby providing access to previously undruggable proteins (also known as PROteolysis TArgeting Chimera [PROTAC] technology) (Pettersson & Crews, 2019). To evaluate acute degradation of the STAG1 protein as a therapeutic concept and mimic activities of a potential STAG1-targeted degrader, we used the auxin-inducible degron (AID) system. Auxin (indole-3-acetic acid; IAA) mediates the interaction of

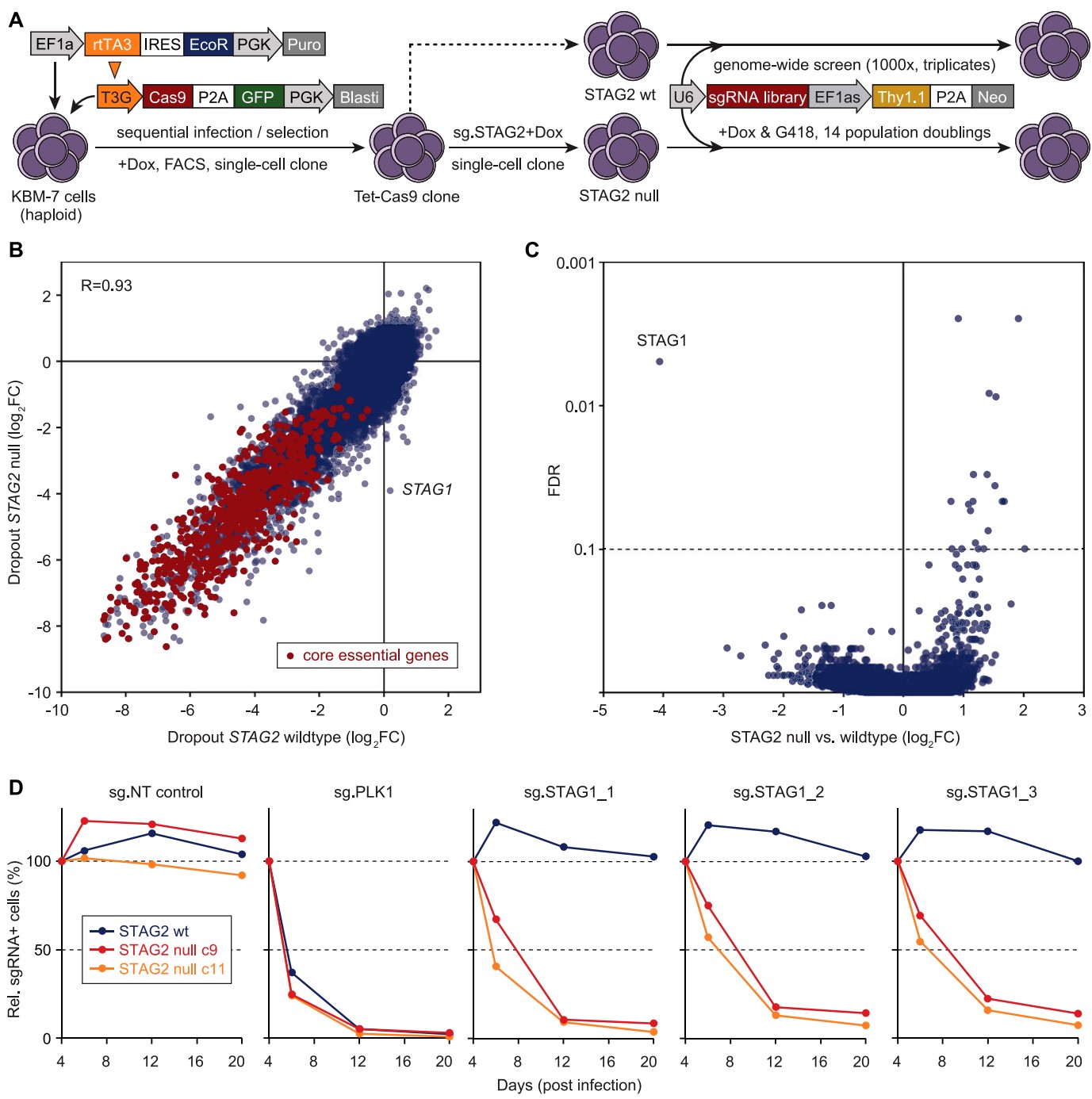

**Figure 1. Genome-wide CRISPR screens in isogenic KBM-7 cell lines identify STAG1 as the most selective synthetic-lethal dependency in *STAG2*-mutant cells.**
**(A)** Schematic of cell line engineering and genome-wide CRISPR screening. Haploid KBM-7 cells were sequentially transduced with indicated lentiviral vectors, Cas9-GFP–expressing cells were single-cell sorted, and derived clones characterized for homogenous and effective CRISPR editing. The selected clone was lentivirally transduced with a validated sgRNA-targeting *STAG2*, and several subclones were isolated and characterized for *STAG2* knockout. One *STAG2* knockout clone was selected ("c9") and screened side-by-side with the parental *STAG2*–wild-type clone ("B4") using a second-generation genome-wide sgRNA library. **(B)** Gene-level dropout effects in wild-type and *STAG2*-null KBM-7 cells. Shown are log₂ fold changes between the end point of triplicate screens and the sgRNA library (analyzed using MAGeCK v0.5.8). A set of previously described generally essential genes (Wang et al, 2019) are highlighted in red. **(C)** Analysis of differential effects in *STAG2*-null versus *STAG2*–wil-type cells (MAGeCK v0.5.8), revealing STAG1 as the most prominent and only significant synthetic lethal interaction. **(D)** Competitive proliferations assays in *STAG2*–wild-type (B4, blue lines) and *STAG2*-mutant (c9, red lines) cell lines used in the screen and in an independent *STAG2*-mutant clone (c11, orange lines). Cells were transduced with a lentiviral vector co-expressing mCherry and the indicated sgRNAs, and the fraction of mCherry+ cells was monitored over time using flow cytometry.

an AID degron domain in a hybrid AID protein with the substrate recognition domain of a transgene-encoded TIR1 E3 ligase. This results in ubiquitylation of the hybrid target protein by recruitment of an SCF-type ubiquitin ligase (E3) and finally in proteasomal degradation (Hayashi, 2012). Using CRISPR/Cas9 genome engineering, we generated HCT 116 colon carcinoma cells that stably integrated a GFP-tagged AID (Morawska & Ulrich, 2013) at the N terminus of both alleles of the endogenous *STAG1* gene in the parental *STAG2*–wild-type cells and an isogenic HCT 116 cell line carrying a previously introduced *STAG2* T220fs null mutation (*STAG2*-502c4, described in van der Lelij et al [2017]) (Fig 2A). Homozygous integration of GFP-AID at the *STAG1* locus was confirmed by PCR genotyping and immunoblotting in two independent clones each for the parental HCT 116 and STAG2-502c4 background (Figs 2B and S3A and B). The four clones were subsequently transduced and selected for TIR1 transgene expression. 48 h after addition of auxin, GFP-AID-STAG1 protein levels were strongly reduced, indicating potent degradation of the hybrid protein (Figs 2B and S3B and C). Degradation of the hybrid protein occurred within 1–2 h after addition of auxin (Fig S3D). Immunofluorescence staining revealed that the degradation of GFP-AID-STAG1 caused mitotic arrest after 48 h of auxin treatment in cells lacking STAG2 yet had no effect on the mitotic index in the parental cell line (Fig 2C). Strikingly, STAG1 degradation elicited severe chromosome alignment defects (Fig 2C lower panel) and a complete loss of sister chromatid cohesion (Fig 2D) in *STAG2*-mutated cells, whereas sparing wild-type counterparts. Consistent with these observations, live-cell imaging revealed an accumulation of mitotic cells before cell death, or abnormal mitotic exit, in *STAG2*-mutated but not wild-type cells upon auxin treatment (Videos 1 and 2 and Fig S3E). Importantly, GFP-AID-STAG1 degradation induced by long-term treatment with auxin greatly reduced the viability of *STAG2*-mutated cells, whereas having no discernible effect on STAG2-proficient parental cells (Fig 2E and F). These results demonstrate that targeting STAG1 by pharmacologically induced degradation selectively abrogates sister chromatid cohesion and cell viability in *STAG2*-mutated cells, without affecting *STAG2*–wild-type cells. Thus, STAG1-directed small molecule degraders that engage a suitable human E3 ligase such as VHL or CRBN may provide a promising therapeutic modality for the treatment of *STAG2*-mutated cancers.

## Determination of the structure of STAG1 segments complexed with RAD21 peptides

We sought to obtain insight into the structure of STAG1 by X-ray crystallography to enable the discovery of STAG1-directed small molecules and to explore additional ways to interfere with STAG1's function. As the STAG2 structure has been determined previously (Hara et al, 2014) and given the high sequence similarity between STAG1 and STAG2 (81% similarity and 70% identity), a dragon-shaped structure with RAD21 interactions spanning nearly the entire length of STAG1 was predicted. The STAG2 structure consists of 17 tandemly arranged HEAT repeat domains complemented at both the N- and C terminus with additional helical elements. The RAD21–STAG2 interface is formed over some 70 residues of RAD21 and was characterized to include four major contact areas, termed sites I–IV (Hara et al, 2014) (Fig 3A). We have designed constructs of STAG1's N-terminal and central region encompassing interaction

regions I and II, and IV, respectively, and obtained crystal structures in both the presence and absence of the corresponding RAD21 peptides (Fig 3A).

We determined the structures of the N-terminal region of human STAG1, encompassing residues 86–420, in the absence and presence of a RAD21 peptide to 2.0 and 2.4 Å, respectively. As expected based on the degree of sequence homology (73% identities over equivalent regions), the structure is highly similar to STAG2 (~1.3 Å RMSD) (Fig 3A). The STAG1–RAD21 peptide complex was obtained with a 25-residue peptide that corresponds to residues 321–345 of human RAD21. The peptide is predominantly in an extended conformation with a single $\alpha$-helix formed at the C terminus and contacts STAG1 around the C-terminal ends of helices in HEAT repeats R1–R4 creating a long, relatively flat interface spanning over 40 Å in length and 1,000 $\text{Å}^2$ in the contact area (sites I and II). The interface is primarily polar, with nine hydrogen bonds and five salt bridges formed between RAD21 and STAG1 (Fig 3B). Most of these contacts are formed in the extended N-terminal half of the RAD21 peptide and are generally of the form: STAG1 side chain to RAD21 side chain (Fig 3C). In contrast to interactions at the N-terminal half of the RAD21 peptide that are very well conserved, the helical section at the C-terminal half of the RAD21 peptide is amphipathic in nature and forms van der Waals contacts with a hydrophobic patch in the STAG1 surface that differs significantly from that observed in the STAG2 complex structure (Fig S4).

The STAG1 central region consists of HEAT repeats R6-R14 (residues 459–915), and the structure has been determined both alone and in complex with a 40–amino acid residue RAD21 peptide (spanning residues 356–395) to 2.3 and 3.1 Å, respectively (site VI, Fig 3A). As is the case for the N-terminal region, the STAG1 structures are very similar to the STAG2 structure (~1.4 Å RMSD), with no significant changes observed upon binding to the RAD21 peptide. The RAD21 peptide adopts a compact conformation, forming two sections of an $\alpha$-helix linked by an extended loop, and interacts with STAG1 within a U-shaped cleft around HEAT repeats R9-R14 (Fig 3A). The site IV interaction interface is extensive (>1,200 $\text{Å}^2$ in area, with 12 hydrogen bonds and 1 salt bridge) and, in contrast to the situation in sites I and II, consists of a mixture of polar and nonpolar interfaces. The two helical regions of the peptide contribute to mostly hydrophobic interactions with buried side chains, whereas the extended loop forms extensive hydrogen-bonded interactions mostly of the form STAG1 side chain to RAD21 main chain (Fig 3D and E). Overall, most of the contacts in site IV are conserved between STAG1 and STAG2. In conclusion, RAD21 interacts with STAG1 through an extended interface that spans nearly the complete length of STAG1.

### STAG1's D797 residue is essential for the interaction with RAD21

We used the structural information obtained to identify STAG1 surface residues that are important for RAD21 binding in an attempt to investigate whether inhibiting STAG1's interaction with the tripartite cohesin ring represents an additional strategy to interfere with STAG1 function. 25 STAG1 surface residues that engage RAD21 were individually mutated to alanine, or to an amino acid of opposite charge, in transgenes encoding FLAG epitope-tagged STAG1 (Fig S5A). Subsequently, STAG1–wild-type and STAG1-mutant variants

were transiently transfected into HEK293 cells and subjected to anti–FLAG-immunoprecipitations. Immunoblotting for cohesin complex subunits demonstrated that wild-type STAG1 and the vast majority of STAG1 mutants efficiently co-immunoprecipitated all core cohesin ring subunits. Importantly, the STAG1 mutations D797K and D797A in site IV strongly reduced the binding to SMC1, SMC3, and RAD21 (Fig S5B and C). At this site, the side chains of STAG1 D797 form hydrogen bonds with the backbone amide residues A377 and Q378 in RAD21. These residues have previously been implicated in interaction with residue D793 in STAG2 (Hara et al, 2014).

Given the strong reduction of cohesin complex binding observed for STAG1 D797 mutants in our cellular co-immunoprecipitation assays, we decided to measure the impact of the mutation in a molecularly defined quantitative real-time assay using surface plasmon resonance. A biotinylated peptide containing RAD21 residues 356–395 was immobilized on a streptavidin-coated sensor chip, and increasing concentrations of either wild-type or D797A-mutant STAG1 protein spanning residues 415–915 were used as analytes. The wild-type STAG1 protein gave a strong dose-dependent response that can be fitted to a 1:1 binding model in kinetic mode, with $K_a = 5.586 \times 10^4 \pm 5.4 \times 10^2$ M$^{-1}$ s$^{-1}$, $K_d = 7.086 \times 10^{-3} \pm 6.6 \times 10^{-5}$ s$^{-1}$, and an equilibrium dissociation constant $K_D = 127$ nM (Fig 3F). Similar parameters ($K_D = 148 \pm 22$ nM) could be determined from a dose–response curve in an equilibrium analysis (Fig 3G). In contrast, the D797A-mutant STAG1 variant showed relatively small responses over the concentration range tested (7–1,000 nM) that appear to be the result of bulk responses (Fig 3F), demonstrating the importance of STAG1's D797 residue in binding RAD21.

### Blocking the interaction of STAG1 with RAD21 as a therapeutic strategy

To explore the cellular consequences of the STAG1 D797 mutations, we generated HCT 116 *STAG2*-502c4 cells stably expressing siRNA-resistant FLAG–STAG1 wild-type and D797K- or D797A-mutant transgenes and selected clones with similar expression levels (Figs 4A and B and S6A). Co-immunoprecipitation experiments from cell lysates demonstrated that the mutations D797K and D797A abrogated the interaction of the FLAG–STAG1 proteins to the cohesin complex (Figs 4C and S6B). Cell fractionation experiments showed that wild-type FLAG–STAG1 protein was enriched in the chromatin-bound fraction. In contrast, both D797K and D797A were depleted from the chromatin-bound fraction and were present almost exclusively in the nuclear soluble fraction (Figs 4D and S6C). These data indicate that the interaction of STAG1 with RAD21 is required for the association of STAG1 with chromatin.

To test whether STAG1 D797 mutations affect cohesin function in cells, we depleted endogenous STAG1 using siRNA in *STAG2*-mutated clones expressing siRNA-resistant wild-type and mutant STAG1 variants. Depletion of endogenous STAG1 in a *STAG2*-mutated background resulted in a strong increase in phospho-Ser10 histone H3–positive mitotic cells and abnormal chromosome alignment in cells transduced with the empty vector (Figs 4E and S6D). These phenotypes could be rescued by the expression of wild-type FLAG–STAG1. Strikingly, the D797K FLAG–STAG1 mutant showed an increased mitotic arrest and decreased chromosome alignment

comparable with empty vector, whereas D797A gave an intermediate phenotype. In accordance with these findings, the D797K STAG1-mutant variant failed to support sister chromatid cohesion upon depletion of the endogenous counterpart, whereas the D797A-mutant partially lost its ability to mediate sister chromatid cohesion in most chromosome spreads (Fig 4F). Whereas wild-type FLAG–STAG1 was able to rescue cell viability upon depletion of endogenous STAG1, the STAG1 D797K-mutant transgene displayed reduced viability (Fig 4G). The D797A mutant displayed intermediate viability, in line with the partial loss of sister chromatid cohesion. Our analyses suggest that the mutation of STAG1 residue D797 renders the protein unable to bind to cohesin, associate with chromatin, support sister chromatid cohesion, and ensure cell viability. Protein–protein interaction inhibitors blocking STAG1's interaction with the RAD21 subunit of the cohesin complex around residue D797 might, therefore, represent an additional promising modality for attacking *STAG2*-mutant tumors.

## Discussion

Discovery of highly prevalent *STAG2* mutations has been an unexpected finding emerging from large-scale cancer genome studies over the last decade that opens up possibilities for the development of new therapeutic strategies and targets. Remarkably, *STAG2* is one of only 12 genes that are significantly mutated in four or more major human malignancies (Lawrence et al, 2014). Most somatic STAG2 mutations are deleterious loss-of-function alterations (Hill et al, 2016). They are thought to occur early during carcinogenesis and are hence likely to be shared by most if not all cells in myeloid neoplasms, bladder cancers, and Ewing sarcoma (Balbas-Martinez et al, 2013; Kon et al, 2013; Yoshida et al, 2013; Crompton et al, 2014; Thol et al, 2014; Thota et al, 2014; Tirode et al, 2014). The recurrent, deleterious, and likely truncal nature of STAG2 mutations strongly suggests that STAG2 loss-of-function alterations represent cancer driver events. This makes STAG2 loss an attractive patient selection biomarker, which if targeted by precision medicine might allow for the eradication of most tumor cells. In this study, we have identified STAG1 as the strongest and most selective dependency of *STAG2*-mutated cells in a genome-wide synthetic lethality screen and validated the dependency in vivo. CRISPR or RNAi-mediated STAG1 inactivation selectively kills *STAG2*-mutant cells but has no effect on STAG2-proficient cells, suggesting a potentially large therapeutic window.

In addition to our screen in isogenic KBM7 cells, genome-wide essentialome screens in hundreds of cancer cell lines and several non-transformed cell types suggest that STAG1, in contrast to many other targets of established cancer therapeutics, is not required for proliferation and survival of the vast majority (~95%) of analyzed cell lines (Meyers et al, 2017; Dempster et al, 2019). Although these results implicate STAG1 as a promising target for selective eradication of *STAG2*-mutant cancer cells, STAG1 has been shown to be essential for embryonic development (Remeseiro et al, 2012), potentially because of gene regulatory functions that remain poorly understood. The overall safety of STAG1 suppression in an adult

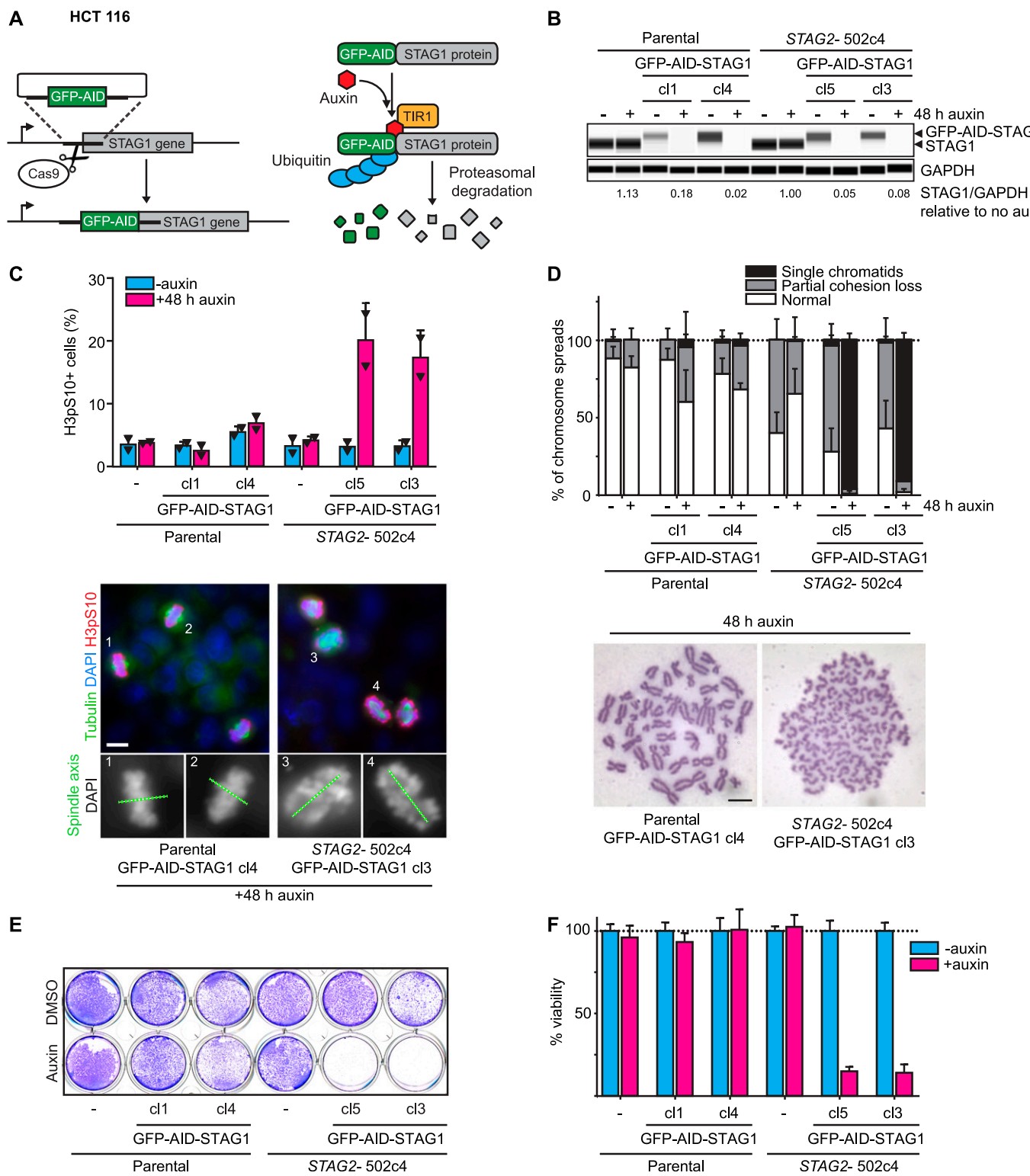

**Figure 2. Auxin-induced degradation of STAG1 causes severe mitotic defects, abrogates sister chromatid cohesion, and inhibits proliferation in *STAG2*-deficient cells.**
**(A)** Parental and *STAG2*-deficient 502c4 HCT 116 cell lines were engineered by N-terminally tagging the endogenous *STAG1* gene with a GFP-AID degron as depicted in the scheme. Auxin mediates the interaction of GFP-AID-STAG1 hybrid protein with ectopically expressed TIR1, leading to ubiquitylation of STAG1 by an E3 ubiquitin ligase, and resulting in proteasomal degradation. **(B)** Capillary immunodetection assay was used to detect and quantify (GFP-AID-) STAG1 protein levels as compared with GAPDH loading control upon 48 h of auxin addition. Quantification of STAG1 protein levels upon auxin addition relative to no auxin is depicted (respective quantification and Western blot in Fig S3B and C). **(C)** Immunofluorescence analysis was performed 48 h after the addition of auxin to determine the mitotic index by scoring the fraction of histone H3 phosphoSer10–positive (H3pS10+) cells (upper panel; n ≥ 584 cells, triangles denote values of two independent experiments, and error bars denote SD) and to

organism cannot be estimated at this point and should be in-vestigated in parallel to drug development efforts.

The absence of an enzymatic activity and lack of a precedence of successfully drugging HEAT repeat proteins suggest that STAG1 may be hard to target in a conventional way. Here, we describe two potential therapeutic strategies to inhibit STAG1. Pharmacologically induced protein degradation of STAG1 as well as abrogation of STAG1's interaction with the RAD21 subunit of cohesin is selectively detrimental to sister chromatid cohesion and viability of *STAG2*-mutated cells. We provide crystal structures of STAG1 segments bound to RAD21 as well as biophysical interaction assays for the two proteins, which can form the basis to identify chemical matter to translate the aforementioned therapeutic approaches into practice.

To maximize the therapeutic window of future STAG1 targeting agents, such as STAG1 degraders or STAG1:RAD21 interaction in-hibitors, selectivity against STAG2 is paramount. This will be challenging given the high similarity between the two paralogs. The STAG1 crystal structures reported here can support the identifi-cation of differences in surface residues that can be exploited to confer paralog selectivity for small-molecule ligands. Notably, targeted degraders against p38 MAPK family and BET-family pro-teins have demonstrated the ability to discriminate between closely related paralogs (Zengerle et al, 2015; Bondeson et al, 2018; Burslem et al, 2019). It is likely that the selectivity in the reported bifunctional degraders is not derived from the binding properties of the target-engaging moiety but rather from paralog-selective as-pects of complex formation with the E3 ligase. Thus, specificity introduced at the level of E3 complex formation may help dis-criminate between STAG1 and STAG2 degradation.

Altogether, our work on STAG1 vulnerabilities and protein structure is essential for the next steps toward transforming a hard wired synthetic lethality into a drug that can be used to treat the estimated half a million patients diagnosed with *STAG2*-mutant malignancies worldwide each year.

# Materials and Methods

### Antibodies

The following antibodies were used: rabbit anti-STAG1 (GTX129912; Genetex), goat anti-STAG2 (A300-158A; Bethyl Laboratories), mouse anti-β-actin (AC-15) (A3854; Sigma-Aldrich), mouse anti-GAPDH (ab8245; Abcam, Fig 2), rabbit anti-GAPDH (ab9485; Abcam, Fig 4), rabbit anti-H3pS10 (06570; Millipore), FITC-conjugated mouse anti-tubulin (F2168; Sigma-Aldrich), rabbit anti-SMC1 (A300-055A; Bethyl Laboratories), mouse anti-SMC3 (ID 646 in HEK293 co-immunoprecipitation immunoblot; Peters Laboratory), rabbit anti-SMC3 (A300-060A in capillary immunodetection assay; Bethyl

Laboratories), mouse anti-RAD21 (05-908; Millipore), mouse anti-tubulin (T5168; Sigma-Aldrich), mouse anti-FLAG (1042E; Sigma-Aldrich, Fig S5), rabbit anti-FLAG (F7425; Sigma-Aldrich, Fig 4), rabbit anti-histone H3 (4499; Cell Signaling), mouse anti-p53 (OP43; Calbiochem), and secondary rabbit (1706515; Bio-Rad), mouse (1706516; Bio-Rad), and goat (P0160; Dako) anti-IgG-HRP.

### Cell culture and cell line engineering

Lenti-X (632180; Clontech), UM-UC-3, and HEK293 cells were maintained in DMEM, RT-112 cells in RPMI 1640, and KBM-7 cells in IMDM medium, all supplemented with 10% FBS, 1 mM sodium py-ruvate, 4 mM L-glutamine, and penicillin/streptomycin (all Invi-trogen). HCT 116 *STAG2*-505c1 and 502c4 cells were described and characterized in van der Lelij et al (2017) and cultured in McCoy's 5A w/glutamax medium supplemented with 10% FBS (both Invitrogen). KBM-7 cells were sequentially transduced with pWPXLd-EF1A-rtTA3-IRES-EcoRec-PGK-Puro (pWPXLd-RIEP) and pSIN-TRE3G-Cas9-P2A-GFP-PGK-Blast and bulk selected for viral integration. A clone, "B4," was derived using single-cell FACS and tested for Cas9 induction and functionality upon treatment with 0.1 μg/ml doxycycline (DOX; Sigma-Aldrich). To generate isogenic STAG2-null sister cell lines, the B4 clone was (co-)transduced with lentiviral vectors (pRRL-U6-sgRNA-EF1as-eBFP2) expressing different sgRNAs targeting STAG2. After 8 d of culture in DOX to complete Cas9-mediated genome editing, single-cell–derived clones were isolated using FACS, ex-panded, and analyzed for STAG2 mutations using Sanger se-quencing and STAG2 expression using immunoblotting (Fig S1A), as well as for Cas9 inducibility and function. For further analyses, we selected two clones harboring frameshift mutations in STAG2 (Table S3), "c9" (generated using expression of sgSTAG2_GTTTC-GACATACAAGCACCC) and "c11" (generated using co-expression of sgSTAG2_GATTTGAACTTCTTCCACTG and sgSTAG2_GGAAAACGAGC-CAATGAG). Before genome-wide screening and validation studies, all three clones (B4, c9, and c11) were sorted for haploid cells using Hoechst 33342 staining. To engineer N-terminal GFP-AID-STAG1 tags at the endogenous *STAG1* locus of HCT 116 parental and *STAG2*-502c4 cell lines, we performed CRISPR/Cas9–mediated genome editing as previously described (Wutz et al, 2017). In short, cells were triple-transfected with Lipofectamine 2000 according to the manufacturer's instructions (Invitrogen) with one construct con-taining STAG1 homology arms and coding sequences for mono-meric EGFP (L221K) and the *Arabidopsis thaliana* IAA1771-114 (AID*) mini-degron and two PX335 constructs containing gRNA1:CACCGA-CAATACTTACTGTAACAC and gRNA2:CACCGTATTTTTTAAGGAAAATTT. Single clones were selected based on GFP positivity using flow cytometry and confirmed to be homozygous by PCR. Two of these clones per genotype were transduced with a lentivirus co-expressing OsTIR1 and a puromycin resistance marker (pRRL-SOP [Muhar et al, 2018]) and subsequently selected using puromycin

---

investigate mitotic spindle geometry and chromosome alignment (lower panel). Scale bar 20 μm. **(D)** Analysis of prometaphase chromosome spreads after auxin addition for 48 h. The status of sister chromatid cohesion of individual metaphase spreads was categorized into normal, partial loss of cohesion, or single chromatid phenotypes (n = 100 spreads, error bars denote SD of at least three independent experiments). Scale bar: 10 μm. **(E, F)** Cell viability was assessed 7 d after auxin or DMSO treatment by crystal violet staining and using a metabolic assay (F, n = 3 biological repeats with each three technical repeats, error bars denote SD). Source data are available for this figure.

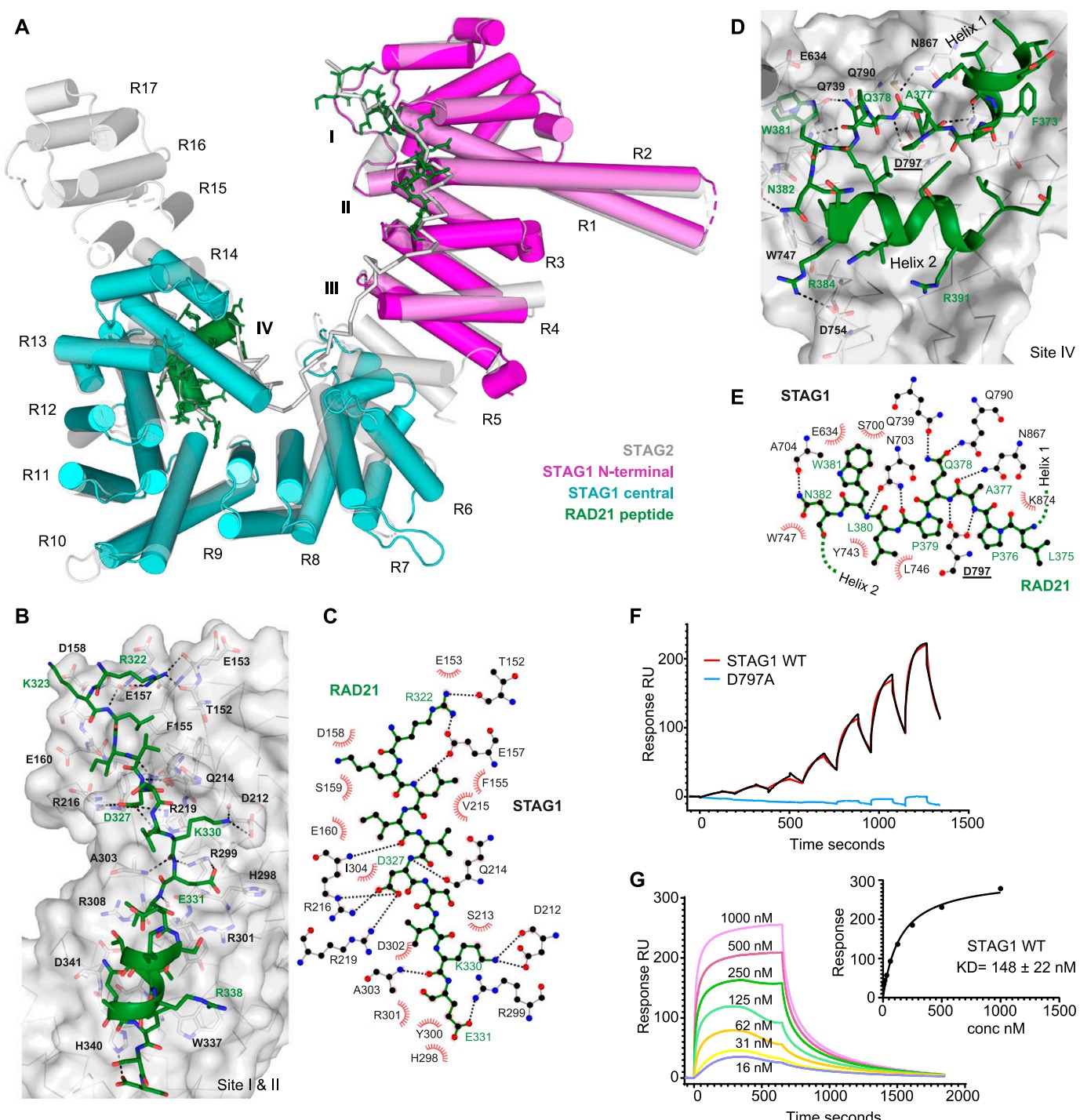

**Figure 3. Structure of STAG1 hotspot regions in complex with RAD21 peptides.**
**(A)** Cartoon diagram of the overall structure of STAG1 N-terminal region (pink), and central region (cyan) superposed on to the STAG2 structure (gray with semi transparency), with RAD21 peptide regions in green and heat repeats numbered. **(B)** Close-up view of the interaction between RAD21 and STAG1 N-terminal region (sites I and II), with hydrogen bonds shown as dashed lines and key interacting residues shown in stick format. **(B, C)** Schematic view of the interaction between RAD21 residues 322-221 and the N-terminal region of STAG1; the orientation and coloring correspond roughly to what is shown in panel (B). Hydrogen bonds are shown as dashed lines and van der Waals contacts are shown as red-spoked arcs. **(D)** Close-up view of the interaction between RAD21 and STAG1 central region (site IV), with hydrogen bonds shown as dashed lines and key interacting residues shown in stick format. The first 12 residues of the RAD21 peptide, which form a further short section of α-helix in the STAG2 structure are disordered in the electron density and have not been included in the model. **(D, E)** Schematic view of the key interactions between RAD21 and STAG1 central region, highlighting the interacting region spanning RAD21 residues 375–382, the orientation and coloring correspond roughly to what is shown in panel (D). Hydrogen bonds are shown as dashed lines, and van der Waals contacts are shown as red-spoked arcs. **(F)** Comparison of STAG1 central region WT versus D797A mutant binding to a RAD21[356–395] peptide by surface plasmon resonance. Biotinylated RAD21[356–395] was immobilized on the chip, and increasing concentrations of STAG1 were

(0.5 μg/ml). Primers used for genotyping were as follows: forward primer: TTTGCTGCATTGTGAAAGGACC and reverse primer: ACTA-TAAGGGGCCCTCCAAAC. HCT 116 cell lines expressing FLAG–STAG1 transgenes were generated by lentiviral transduction by use of Lenti-X Packaging Single Shots (VSV-G) one shot LentiX kit (Clontech) followed by puromycin selection (2 μg/ml). All cell lines were tested negatively for mycoplasma contamination and have been authenticated by STR fingerprinting. Sources, STAG2 status, and authentication information of cell lines used in this study are provided in Table S3.

### Human genome-wide sgRNA library

The design, construction, and performance of the human genome-wide Vienna sgRNA library are described in detail in Michlits et al, 2020. In brief, the library was designed to target 18,659 human RefSeq genes with six sgRNAs per gene (Table S1), which were selected based on positioning in the 5′ region of coding sequences, the presence of a natural G within the first three nucleotides at the 5′-end, and a nucleotide composition score that was derived based on re-analyses of early CRISPR/Cas9–based dropout screens. After excluding sgRNAs targeting multiple coding genes, the 5′-ends of 20mer sequences were trimmed to the G, resulting in a library of 18-, 19-, and 20-mer sgRNAs that harbor a natural G at the 5′-end. To construct the library, these sequences were flanked by primer-binding and BsmBI restriction sites, synthesized on a 244K oligonucleotide array (Agilent Technologies), and cloned as 13 sub-pools into the lentiviral sgRNA expression vector sgETN (pLenti-U6-sgRNA-EF1as-Thy1.1_P2A_NeoR). All sub-pools were constructed using a high representation (>5,000× bacterial colonies per sgRNA) and pooled in an equimolar manner to create the final library.

### Lentiviral transduction and genome-wide CRISPR screening

Pooled CRISPR library virus production was performed by standard transient transfection of Lenti-X cells in 10-cm dishes. Per dish, 1 × 10$^7$ cells were seeded and 8 h later transfected with a DNA mix comprising 800 μl of 150 mM NaCl, 4 μg sgETN library, 2 μg pCMVR8.74 (# 22036; Addgene), and 1 μg pMD2.G (# 12259; Addgene) and mixed with 21 μl of PEI (1 mg/ml Stock, 25K linear, 333; Polysciences). Before transfection, the DNA mix was briefly vortexed and incubated for 25 min at room temperature followed by dropwise addition to the packaging cells. The next day, the cells were supplemented with fresh medium, and viral supernatant was harvested 48 and 72 h posttransfection. Both harvests were pooled and filtered through a 0.45-μm PES filter (VWR) and stored at 4°C until infection of the target cells. For the genome-wide STAG2 synthetic lethal screen, at least 1.2 × 10$^9$ STAG2-null or wild-type haploid KBM-7 cells were infected in the presence of 4 μg/ml polybrene (Merck Millipore) by adding filtered viral supernatant to the cell suspension at 2 × 10$^6$ cells/ml. Each condition was maintained in three

replicates with a library representation of 500×. 4 d postinfection, the transduction efficacy was determined using antibody staining for Thy1.1 (APC anti-mouse CD90.1, 202526; BioLegend) and flow cytometry analysis. Upon confirming an MOI of <0.2 to ensure single viral integration, the cells were selected with 1 mg/ml G418 (Gibco), and Cas9 expression was induced by adding 0.1 μg/ml DOX. After 28 d of routine passaging, genomic DNA (gDNA) was extracted and processed for next generation sequencing (NGS). To transduce individual sgRNAs for validation studies, 1 × 10$^6$ Lenti-X were seeded per well of a six-well plate 8 h before transfection. 12 μl PEI was mixed with 2 μg Lentiviral plasmid, 1 μg pCMVR8.74, and 0.5 μg pMD2.G in 200 μl serum-free media and incubated for 25 min at RT before adding to the cells. Viral supernatant was harvested 48 and 72 h posttransfection, filtered, and used for infection of the target cells in the presence of 4 μg/ml polybrene.

### gDNA extraction and NGS library preparation

NGS libraries were prepared from gDNA extracted from day 28 samples as previously described (Rathert et al, 2015). Briefly, the cells were washed with PBS, lysed, incubated with Proteinase K, and gDNA-purified by two rounds of phenol extraction and subsequent EtOH precipitation. For NGS library generation, two sequential PCR reactions were performed. The first PCR amplifies the sgRNA with primers introducing a sample barcode and partial Illumina sequencing adaptors, which were filled up to a complete Illumina adaptor sequence with the second PCR. The first PCR contained, in each 50 μl PCR reaction, a mixture of 1 μg gDNA, 5 μl 10× PCR-buffer, 4 μl MgCl$_2$ (25 mM), 1 μl dNTP (10 mM each), 1.5 μl forward primer (10 μM), 1.5 μl reverse primer (10 μM), and 0.2 μl AmpliTaq Gold (#4311820; Invitrogen). Thermocycler conditions were as follows: 10 min at 95°C, 28 cycles of 30 s at 95°C, 45 s at 57°C, 30 s at 72°C, and final 7 min at 72°C (forward primer: 5′-GCATACGAGATAGCTAGCCACC-3′; reverse primer: 5′-CTCTTTCCCTACACGACGCTCTTCCGATCTNNNNNNNXXXXTTCCAGCA-TAGCTCTTAAAC-3′, where *NNNNNN denotes randomly synthesized nucleotides and XXXX denotes sample barcodes). The amplicon was concentrated via silica columns and gel-purified before it was tagged with primers containing the rest of the standard Illumina adaptors. The second PCR was performed as described above with the only difference that each reaction contained 10 ng DNA-purified amplicon per 50 μl reaction and the following thermocycler conditions: 10 min at 95°C, four cycles of 30 s at 95°C, 45 s at 62°C, and 30 s at 72°C; with 7-min final extension at 72°C (forward primer 2: AATGATACGGCGACCACCGAGATCTACACTCTTTCCCTACACGACGCT; reverse primer 2: CAAGCAGAAGACGGCATACGAGATAGCTAGCCACC).

### Animal studies, shRNAs, and competition assays

All animal experiments were performed according to project licenses granted and regularly controlled by the Austrian veterinary authorities. 6–8 wk old, female athymic nude mice were used for the

injected in a single cycle kinetic experiment, a clear dose-dependent binding response is observed for the wild-type protein (red curve, with fit to a 1:1 binding model shown as black line), whereas only small nonspecific responses are seen for the D797A variant (blue curve). **(G)** Analysis of the STAG1 central region–RAD21[356–395] interaction on surface plasmon resonance using equilibrium analysis. Increasing concentrations of STAG1 were injected over the sensor surface, and the response at equilibrium is fit to a dose–response curve (inset) with an apparent K$_d$ of 148 ± 22 nM.

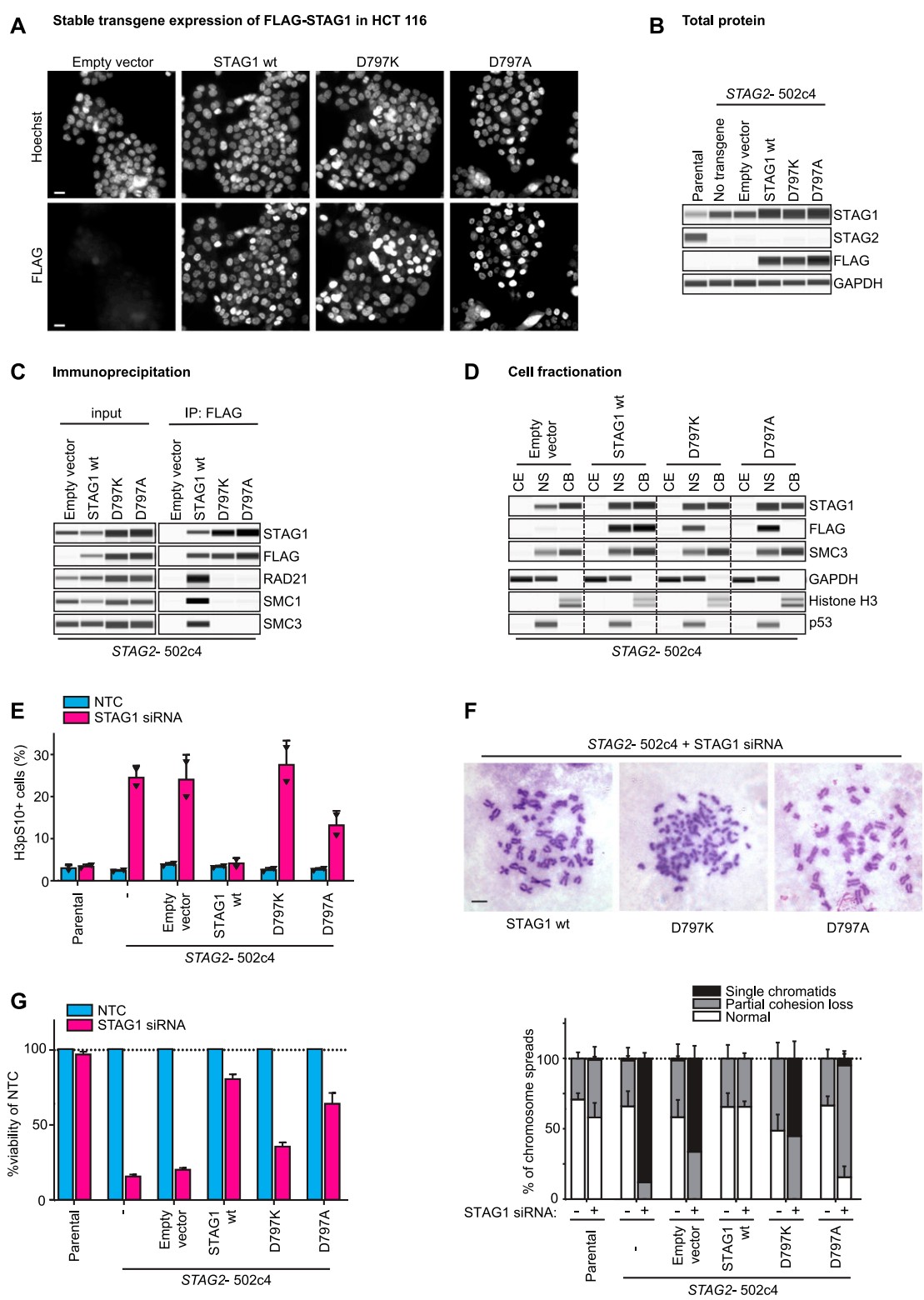

**Figure 4. STAG1 D797 residue is essential for binding to the cohesin ring, chromatin localization, sister chromatid cohesion, and cell viability in *STAG2*-deficient cells.**
**(A)** HCT 116 *STAG2*-502c4 cells were stably transduced with FLAG-tagged, siRNA-resistant wild-type (wt) and mutant STAG1 or empty vector control transgenes, and nuclear expression was assessed by FLAG immunofluorescence. Scale bar: 20 $\mu m$. **(B)** Capillary immunodetection assay was used to detect FLAG–STAG1 protein levels (see Fig S6A for quantifications). **(C)** Protein extracts from stably expressing FLAG–STAG1 wild-type and mutant cells were subjected to FLAG immunoprecipitation and analyzed for co-precipitation of cohesin ring members by capillary immunodetection assay (see Fig S6B for quantifications). **(D)** Cytoplasmic extract (CE), nuclear-soluble (NS), and chromatin-bound (CB) fractions were obtained to determine the subcellular distribution of FLAG–STAG1 wild-type and mutant protein by capillary immunodetection

animal experiment with the human cell line UM-UC-3. Cells were infected with an rtTA3 expression vector (pRRL-RIEP; pRRL-SFFV-rtTA3-IRES-EcoR-PGK-Puro); selected with puromycin (0.5 µg/ml); subsequently transduced with LT3GEN (pRRL-TRE3G-GFP-miRE-PGK-Neo) harboring sh.Ren.713, sh.STAG1.2076, or sh.STAG1.2949 (Table S4); and selected with G418 (1 mg/ml; Gibco) for 10 d. $1 \times 10^6$ cells were injected subcutaneously in growth factor–reduced Matrigel/PBS (1:1) (BD Biosciences) with one injection site per flank. Five mice were injected for each shRNA resulting in 10 tumors per group. 7 d after injection, the drinking water was supplemented with 2 mg/ml DOX and 5% sucrose. Tumor size was measured with calipers 7, 11, 14, and 18 d postinjection. Tumor volumes were calculated using formula V = (length × width^2)/2. At the end of the experiment, mice were euthanized, tumors were dissected, cells dissociated, and subjected to flow cytometry analyses. For shRNA competition assays in vitro, LT3GEN vectors expressing different shRNAs coupled to GFP were lentivirally transduced into rtTA3-expressing cell lines. Knockdown and, thus, GFP expression was induced with 1 µg/ml DOX treatment. The fraction of GFP-positive cells was determined at indicated time points using an IQue flow cytometer (Intellicyt) and normalized to the first measurement and control shRNAs. All shRNA sequences are provided in Table S4.

### Auxin induced degradation, cell viability, crystal violet assay, and siRNA transfection

For AID experiments, the cells were incubated in the presence of DMSO or 500 µM auxin (I5148; Sigma-Aldrich), which was refreshed every 2 d. Viability was determined using CellTiter-Glo (Promega), and by staining with crystal violet (HT901; Sigma-Aldrich). For knockdown experiments, the cells were transfected with ON-TARGETplus SMARTpool siRNA duplexes (catalog ID L-010638-01; Dharmacon) and the Lipofectamine RNAiMAX reagent according to the manufacturer's instructions (Invitrogen). Chromosome spreads, capillary immunodetection assay, immunoblotting, and immunofluorescence experiments were performed using a final siRNA concentration of 20 nM. Cell viability assay was performed using 10 nM siRNA.

### Cell extracts for capillary immunodetection assay, immunoblotting, FLAG-immunoprecipitation, and cell fractionation

For capillary immunodetection assay and immunoblotting in HCT 116 GFP-AID-STAG1 and FLAG-immunoprecipitation in HEK293, the cells were resuspended in a solution containing 25 mM Tris–HCl, pH 7.5, 100 mM KCl, 5 mM MgCl$_2$, 0.2% NP-40, 10% glycerol, 1 mM NaF, complete protease inhibitor mix (Roche), and benzonase (VWR), and HCT 116 stable transgene cell lines were resuspended in a solution containing 50 mM Tris–Cl, pH 8.0, 150 mM NaCl, 1% Nonidet

P-40 supplemented with complete protease inhibitor mix (Roche), and phosphatase inhibitor cocktails (P5726 and P0044; Sigma-Aldrich) and lysed on ice. For co-immunoprecipitation in HCT 116 stable transgene cell lines, the cells were resuspended in (50 mM Tris–Cl, pH 7.4, 150 mM NaCl, and 0.5% NP-40). For FLAG-STAG1 co-immunoprecipitation, lysates were spun down for 10 min, followed by FLAG-immununoprecipitation using anti-FLAG M2-Agarose Affinity Gel (Sigma-Aldrich) for 2 h (HEK293) or o/n (HCT 116 stable transgene cell lines) and washing with lysis buffer. Input lysates and immunoprecipitates were resuspended in SDS sample buffer and heated to 95°C. Capillary immunodetection assay was performed according to the manufacturer's instructions (WES; Protein Simple) and analyzed using Compass for Simple Western software. For subcellular protein fractionation, the cells were spun down and lysed stepwise according to the manufacturer's instructions (Pierce).

### Immunofluorescence, live cell imaging, and chromosome spreads

For immunofluorescence, the cells were fixed with 4% paraformaldehyde for 15 min, permeabilized with 0.2% Triton X-100 in PBS for 10 min, and blocked with 3% BSA in PBS containing 0.01% Triton X-100. The cells were incubated with primary and secondary antibody (Alexa 594; Molecular Probes), DNA was counterstained with Hoechst 33342 (H3570; Molecular Probes) or DAPI (D9542; Sigma-Aldrich), and tubulin was sequentially stained with an FITC-conjugated antibody. Coverslips and chambers were mounted with ProLong Gold (Molecular Probes). Images were taken with an Axio Imager Z2 Stativ microscope and processed with Zen blue software (Zeiss). A Celldiscoverer 7 (Zeiss) automated microscope was used to record live cells for GFP, SiR-DNA (SC007; Spirochrome), and differential interference contrast signals. Data analysis was performed with Microsoft Excel 2013 and GraphPad Prism 8.1.1. For chromosome spread analysis, nocodazole was added to the medium for 45 min–3 h at 100 ng/ml. The cells were harvested and hypotonically swollen in 40% medium/60% tap water for 10 min at room temperature. The cells were fixed with freshly made Carnoy's solution (75% methanol, 25% acetic acid), and the fixative was changed three times. For spreading, cells in Carnoy's solution were dropped onto glass slides and dried. The slides were stained with 5% Giemsa (Merck) for 4 min, washed briefly in tap water, and air dried. For chromosome spread analysis, two independent slides were scored blindly per experiment.

### Cloning, overexpression, and purification of STAG1 for crystallization

STAG1 constructs corresponding to the N-terminal (site I 86-420) and central regions (site IV 459-915) were cloned in the vectors

---

assay (for quantifications, see Fig S6C). **(E)** HCT 116 cells were transfected with nontarget control (NTC) and STAG1 siRNA duplexes, and immunofluorescence analysis was performed 72 h after transfection to determine the mitotic index by scoring the fraction of histone H3 phosphoSer10-positive (H3pS10+) cells (n ≥ 437 cells, triangles denote values of two independent experiments and error bars denote SD). **(F)** Prometaphase chromosome spreads were prepared 72 h after transfection of cells with NTC control or STAG1 siRNA duplexes. The status of sister chromatid cohesion of individual Giemsa spreads was categorized into normal, partial loss of cohesion, or single chromatid phenotypes (n = 100 spreads, error bars denote SD of two independent experiments with each two technical replicates). Scale bar: 10 µm. **(G)** Cell viability was assessed 5–6 d after siRNA transfection using a metabolic assay, and viability was normalized to NTC control (n = 4 biological repeats, error bars denote SD). Source data are available for this figure.

pNIC-ZB (GenBank: GU452710.1) and pNIC28-Bsa4 (GenBank: EF198106.1), respectively, using ligation-independent cloning and transformed into *Escherichia coli* BL21 (DE3)-R3-pRARE2 cells for over-expression. The cells were grown at 37°C in terrific broth medium supplemented with 50 μg/ml kanamycin until an optical density of 2–3 and induced by the addition of 0.3 mM IPTG and incubated overnight at 18°C. The cells were harvested by centrifugation. For purification, cell pellets were thawed and resuspended in buffer A (50 mM Hepes, pH 7.5, 500 mM NaCl, 5% glycerol, 10 mM imidazole, and 0.5 mM Tris [2-carboxyethyl] phosphine [TCEP]), with the addition of 1× protease inhibitor set VII (Merck). Cells were lysed by sonication and cell debris pelleted by centrifugation. Lysates were loaded on to a Ni-Sepharose immobilized metal-affinity chro-matograpy (IMAC) gravity flow column (GE Healthcare), washed with two column volumes of wash buffer (buffer A supplemented with 45 mM imidazole), and eluted with 300 mM imidazole in buffer A. For the N-terminal region, the IMAC elution fraction was im-mediately applied to a 5-ml HiTrap SP Sepharose Fast Flow col-umn (GE Healthcare), washed with two column volumes of elution buffer, and eluted with three column volumes of 50 mM Hepes, pH 7.5, 1 M NaCl, and 5% glycerol. The purification tag was cleaved with the addition of 1:20 mass ratio of His-tagged tobacco etch virus protease during overnight dialysis into buffer A. Tobacco etch virus was removed by IMAC column rebinding, and final protein purification was performed by size-exclusion chromatography using a HiLoad 16/60 Superdex s200 column in buffer A. Protein concentrations were determined by measurement at 280 nm (NanoDrop) using the calculated molecular mass and extinction coefficients. Protein masses were checked by LC/ESI-TOF mass spectrometry. Mutant versions of the central region used in surface plasmon resonance assays were generated from the wild-type constructs by site-directed mutagenesis. Expression and purification were as for the wild type.

### Crystallization and structure determination

For crystallization of the site I crystals, the protein was concen-trated to 15 mg/ml and crystallized by sitting drop vapor diffusion. Crystals grew in conditions containing 0.1 M Na/K phosphate, pH 6.0, 0.2 M NaCl, and 34% PEG200. Initial crystals were substantially improved by seeding using a 1,000 fold dilution of seed stock. Crystals were loop-mounted and cryo-cooled by plunging directly into liquid nitrogen. For crystallization of the site I peptide–bound crystals, an RAD21 peptide corresponding to the sequence KRKLIVDSVKELDSKTIRAQLSDYS was mixed with the protein in a 1:1 ratio, and crystallization trials were set up at 12 mg/ml. The N-terminal region peptide–bound crystals appeared in conditions containing 2.1 M ammonium sulfate and 0.1 M MES, pH 6.3. Crystals were transferred to a cryosolution containing well solution sup-plemented with 25% glycerol before being loop-mounted and plunged into liquid nitrogen. For crystallization of the site IV crystals, the protein was concentrated to 5 mg/ml and crystallized by sitting drop vapor diffusion from conditions containing 20% PEG 3350, 10% ethylene glycol, 0.2 M sodium malonate, and 0.1 M Bis–Tris propane, pH 7.0. Crystals were loop-mounted and cryo-cooled by plunging directly into liquid nitrogen. For crystallization of the site IV peptide–bound crystals, an RAD21 peptide corresponding to the

sequence PTKKLMMWKETGGVEKLFSLPAQPLWNNRLLKLFTRCLTP was mixed with the protein in a 1:1 ratio and crystallization trials were set up at 8.5 mg/ml. Crystals grew from conditions containing 16% PEG 3350, and 0.2 M DL-mallic acid. Crystals were loop-mounted and transferred to a cryosolution containing well solution supple-mented with 20% ethylene glycol before being loop-mounted and plunged into liquid nitrogen. All data were collected at Diamond Light Source beamlines I04-1 (site I, site I peptide, and site IV) and I24 (site IV peptide). Data were processed using DIALS (Winter et al, 2018), and the structures were solved by molecular replacement using the program PHASER (McCoy, 2007) and the structure of the STAG2 RAD21 complex (4PK7) as a search model. A full summary of data collection and refinement statistics can be found in Table S5.

### Analysis of STAG1 RAD21 binding by surface plasmon resonance

A peptide corresponding to RAD21 residues 356–395 was purchased with N-terminal biotin followed by a trioxatridecan–succinamic acid spacer (Biotin-Ttds-PTKKLMMWKETGGVEKLFSLPAQP LWNNRLLKLFTRCLTP). Binding experiments were performed using a Biacore S200 in-strument and a SA sensor chip in a buffer consisting of 10 mM Hepes, pH 7.5, 150 mM NaCl, and 0.5 mM TCEP. ~400 RU of the peptide was immobilized on the chip by injecting for 60 s at 100 nM con-centration. Single cycle kinetic analysis was performed by injecting STAG1 at increasing concentrations (7.5, 16, 31, 62, 125, 250, 500, and 1,000 nM) for 120 s, and equilibrium analysis was performed with a 650-s association and 1,200-s dissociation phase. Both data were fit with Biacore S200 evaluation software, with the final concentration being removed from the fit of the kinetic data for optimal fit (Chi$^2$ = 16.2 RU$^2$, Uvalue = 2).

## Supplementary Information

## Acknowledgements

The IMP is supported by Boehringer Ingelheim and the Austrian Research Promotion Agency (Headquarter grant FFG-852936). P van der Lelij is a member of the Boehringer Ingelheim Discovery Research global post-doc program. Research in the laboratory of J-M Peters is funded by the European Research Council under the European Union's Horizon 2020 research and innovation program (GA 693949) and the Human Frontier Science Program (grant RGP0057/2018). Research in the laboratory of O Gileadi is supported by the SGC, a registered charity (number 1097737) that receives funds from AbbVie, Bayer Pharma AG, Boehringer Ingelheim, Canada Foundation for Innovation, Eshelman Institute for Innovation, Genome Canada, Innovative Medicines Initiative (European Union-EU/European Federation of Phar-maceutical Industries and Associations-EFPIA) (ULTRA-DD grant no. 115766), Janssen, Merck KGaA Darmstadt Germany, MSD, Novartis Pharma AG, Ontario Ministry of Economic Development and Innovation, Pfizer, São Paulo Re-search Foundation-FAPESP, Takeda, and Wellcome (106169/ZZ14/Z). Re-search in the laboratory of J Zuber has received funding from the European Community's Seventh Framework Programme (FP7/2007-2013) ERC StG GA 336860, the European Research Council under the European Union's Horizon 2020 research and innovation program (ERC PoC GA 862507), and the Austrian Science Fund, FWF (GA SFB F47).

## Author Contributions

P van der Lelij: conceptualization, data curation, formal analysis, supervision, investigation, methodology, and writing—original draft, review, and editing.

JA Newman: conceptualization, data curation, formal analysis, investigation, methodology, and writing—original draft, review, and editing.

S Lieb: data curation, formal analysis, investigation, methodology, and writing—review and editing.

J Jude: conceptualization, data curation, formal analysis, investigation, methodology, and writing—original draft.

V Katis: data curation and formal analysis.

T Hoffmann: conceptualization, data curation, and formal analysis.

M Hinterndorfer: data curation and formal analysis.

G Bader: methodology.

N Kraut: resources and writing—review and editing.

MA Pearson: resources and writing—review and editing.

J-M Peters: supervision, funding acquisition, methodology, and writing—review and editing.

J Zuber: conceptualization, supervision, funding acquisition, methodology, and writing—original draft, review, and editing.

O Gileadi: conceptualization, supervision, funding acquisition, methodology, and writing—original draft, review, and editing.

M Petronczki: conceptualization, supervision, and writing—original draft, review, and editing.

## Conflict of Interest Statement

S Lieb, G Bader, N Kraut, MA Pearson, and M Petronczki are full-time employees of Boehringer Ingelheim RCV. All other authors declare no competing interest.

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
