## [Reviewer comments · Life Science Alliance]

Life Science Alliance

STAG1 vulnerabilities for exploiting cohesin synthetic lethality in STAG2-deficient cancers

Petra van der Lelij, Joseph Newman, Simone Lieb, Julian Jude, Vittorio Katis, Thomas Hoffmann, Matthias Hinterndorfer, Gerd Bader, Norbert Kraut, Mark Pearson, Jan-Michael Peters, Johannes Zuber, Opher Gileadi, and Mark Petronczki

DOI: <https://doi.org/10.26508/lsa.202000725>

Corresponding author(s): Petra van der Lelij, Research Institute of Molecular Pathology (IMP); Johannes Zuber, IMP Vienna; and Opher Gileadi, University of Oxford

Review Timeline:	Submission Date:	2020-03-31
	Editorial Decision:	2020-04-29
	Revision Received:	2020-05-13
	Accepted:	2020-05-14

Scientific Editor: Andrea Leibfried

Transaction Report:

April 29, 2020

RE: Life Science Alliance Manuscript #LSA-2020-00725-T

Dr. Petra van der Lelij
Research Institute of Molecular Pathology (IMP)
Campus Vienna Biocenter 1
Vienna, Vienna 1030
AUSTRIA

Dear Dr. van der Lelij,

Thank you for submitting your manuscript entitled "STAG1 vulnerabilities for exploiting synthetic lethality in STAG2-deficient cancers" to Life Science Alliance. Your manuscript has been now assessed by two reviewers, and I append their reports below.

As you will see, both reviewers appreciate your analysis and are supportive of publication here, pending minor revision. We would thus like to invite you to submit a final version to us, addressing the comments made by the reviewers. It is fine to respond to the first comment of rev#3 in your point-by-point response and by changes to the manuscript text, because you already have a small evaluation of the patient-associated mutant STAG2 p.K983*. Some editorial issues need addressing, too:

- please check your figure legends / manuscript text carefully - not all clones used are properly described, making it mandatory to read your previous papers to understand the current manuscript fully. Figures, figure legends and manuscript text should be understandable without going through the existing literature, please.
- please check your M&M section carefully, not all experimental procedures are described; importantly, the mouse work and associated ethics committee approval is not mentioned => please see our guidelines to include a proper statement; siRNA/shRNA description is lacking for example as well
- please indicate actual p-values in either figure or figure legend whenever statistical tests have been employed (only *** shown at the moment)
- please fill in all mandatory fields in our submission system when submitting the revised version (the system will prompt you to do so)
- please make sure that the corresponding author(s) listed in our submission system match those listed as such in the manuscript file; please have all corresponding authors link their ORCID iD to their profiles in our submission system
- please include a COI statement into your manuscript text
- please upload the missing movie, and re-name it to video 1
- please note that we have only supplementary files, not EV ones (please rename); all figures should get uploaded as individual files (also suppl. figures), all legends should remain in the main manuscript docx file, the tables should remain in excel format but need re-naming, too.

A. FINAL FILES:

B. MANUSCRIPT ORGANIZATION AND FORMATTING:

****Reviews, decision letters, and point-by-point responses associated with peer-review at Life Science Alliance will be published online, alongside the manuscript. If you do want to opt out of having the reviewer reports and your point-by-point responses displayed, please let us know**

immediately.**

Thank you for your attention to these final processing requirements.

Sincerely,

Reviewer #2 (Comments to the Authors (Required)):

In this study Van der Lelij et al. aim to find therapeutic targets for cancers with mutations in STAG2, a tumour suppressor and subunit of the cohesin complex. Using CRISPR-Cas9 screens they found that depletion of its paralog STAG1 resulted in a strong synthetic lethal relationship, confirming previous reports in the literature. The mechanism of lethality is due to a redundant structural role of STAG1 with STAG2 in cohesin complex assembly and sister chromatid cohesion. Importantly, the synthetic lethal relationship was maintained in a STAG2-mutated bladder cancer CDX model indicating a potential for translation into the clinic. STAG1 has no known druggable enzymatic functions so the authors propose two different strategies for disrupting STAG1 in the clinic: Pharmacological induction of STAG1 degradation by PROTAC and disruption of STAG1 physical interaction with the cohesin subunit RAD21. Using the AID-degron system they demonstrate that STAG1 can be efficiently targeted for ubiquitin-mediated proteolysis. They solved the crystal structure of the STAG1 N-terminal region complexed with a RAD21 peptide previously shown to interact with STAG2. A mutational scan of STAG1 on the interaction interface identified STAG1-D797 as an essential residue for physical interaction with RAD21. Expression of D797K or D797A in STAG2 mutant cells was unable to complement the sister chromatid cohesion defects and lethality after depletion of endogenous STAG1, clearly demonstrating that disruption of STAG1-RAD21 interactions has therapeutic potential.

In general, the manuscript is well written and has quality figures that support the authors claims. The work confirms previous work such as the STAG1 and STAG2 synthetic lethal relationship (Van der Lelij Elife 2017, Benedetti 2017 Oncotargets) and CDX model efficacy (Liu 2018 JCI). But the contribution of the STAG1-RAD21 complex structure and binding interface is significant and will aid in development of small molecules to disrupt STAG1-RAD21 physical interaction.

I have no specific issue that needs to be addressed.

Reviewer #3 (Comments to the Authors (Required)):

This manuscript evaluates the genetic dependencies of STAG2 null cells using a genome wide CRISPR screen. STAG1 is identified as the most dependent protein in STAG2 null cells. Using an inducible degron system the authors show that degradation of STAG1 specifically induces cell death by inhibiting sister chromatid cohesion. The interaction with RAD21 and a critical amino acid of STAG1 for this interaction is characterized by resolving the crystal structure of the STAG1-RAD21 interface.

This is an excellent manuscript that provides the basis for developing targeted therapies for STAG2 mutated tumors.

Comments:

1. STAG2 mutations are very heterogeneous and not all may result in a complete loss of the STAG2 activity. The authors should therefore evaluate their approach in primary patient cells with STAG2 mutation.
2. The authors should discuss whether STAG2 mutations are rather driver or passenger mutations and how homogeneously tumor populations are mutated. If the STAG2 mutated clone is a subclone of the tumor a potential treatment may only inhibit a small proportion of the cells.

Response to Referees - LSA-2020-00725-T

Van der Lelij, et al.

We would like to thank the referees for the comments and constructive criticism regarding our work. We are excited about the opportunity to be able to resubmit a revised version of our work addressing the referees' points. In the revised version of our manuscript, we have addressed the specific points raised by reviewer#3 by including additional text and clarifications. Text changes in the revised version of our manuscript compared to the original submission are highlighted by track changes in the manuscript file. A detailed point-by-point response to the comments is listed below.

Reviewer #3 (Comments to the Authors):

This manuscript evaluates the genetic dependencies of STAG2 null cells using a genome wide CRISPR screen. STAG1 is identified as the most dependent protein in STAG2 null cells. Using an inducible degron system the authors show that degradation of STAG1 specifically induces cell death by inhibiting sister chromatid cohesion. The interaction with RAD21 and a critical amino acid of STAG1 for this interaction is characterized by resolving the crystal structure of the STAG1-RAD21 interface.

This is an excellent manuscript that provides the basis for developing targeted therapies for STAG2 mutated tumors.

Comments:

1. STAG2 mutations are very heterogeneous and not all may result in a complete loss of the STAG2 activity. The authors should therefore evaluate their approach in primary patient cells with STAG2 mutation.
2. The authors should discuss whether STAG2 mutations are rather driver or passenger mutations and how homogeneously tumor populations are mutated. If the STAG2 mutated clone is a subclone of the tumor a potential treatment may only inhibit a small proportion of the cells.

We thank the reviewer for bringing up these two important points, which we have included and discussed in detail in our revised manuscript.

Ad 1.) STAG2 mutations are indeed very heterogenous at the DNA level, but to our knowledge most of them are considered to be deleterious based on sequence annotation, as reviewed by Hill et al. 2016 (PMID: 27207471). This notion is further supported by immunohistochemistry studies of biopsies taken from patients with STAG2-mutated bladder cancer and Ewing sarcoma, which revealed no detectable STAG2 protein in the vast majority of cases (PMID: 24121791, 25186949). Moreover, in our previous manuscript (van der Lelij et al. 2017, PMID: 28691904), we have investigated the functional dependency on STAG1 in a panel of bladder cancer and Ewing sarcoma cell lines (both STAG2 positive and negative) and found that all cell lines harboring STAG2 mutations, despite their diversity at the DNA level, are uniformly and selectively sensitive to RNAi-mediated suppression of STAG1. In line with findings of this study, this suggests that the majority of STAG2 mutations are indeed deleterious and a strong predictor for a synthetic-lethal dependency on STAG1.

Ad 2.) STAG2 belongs to a small group of only 12 genes that are significantly mutated in four or more major human malignancies (Lawrence et al., 2014). STAG2 mutations are thought to occur early during carcinogenesis and are hence likely to be shared by most if not all cells in myeloid neoplasms, bladder cancers and Ewing sarcoma (Balbas-Martinez, Sagrera et al., 2013, Crompton, Stewart et al., 2014, Kon, Shih et al., 2013, Thol, Bollin et al., 2014, Thota et al., 2014, Tirode, Surdez et al., 2014, Yoshida, Toki et al., 2013). The recurrent, deleterious, and likely truncal nature of STAG2 mutations strongly suggests that STAG2 loss-of-function alterations represent cancer driver events. This makes STAG2 loss an attractive patient selection biomarker which if targeted by precision medicine might allow for the eradication of most tumor cells. We have included these discussion points in our revised manuscript and thank the reviewer for bringing this important aspect to our attention.

May 14, 2020

RE: Life Science Alliance Manuscript #LSA-2020-00725-TR

Dr. Petra van der Lelij
Research Institute of Molecular Pathology (IMP)
Campus Vienna Biocenter 1
Vienna, Vienna 1030
Austria

Dear Dr. van der Lelij,

Thank you for submitting your Research Article entitled "STAG1 vulnerabilities for exploiting cohesin synthetic lethality in STAG2-deficient cancers". I appreciate the introduced changes and it is a pleasure to let you know that your manuscript is now accepted for publication in Life Science Alliance. Congratulations on this interesting work.

*****IMPORTANT:** If you will be unreachable at any time, please provide us with the email address of an alternate author. Failure to respond to routine queries may lead to unavoidable delays in publication.*******

DISTRIBUTION OF MATERIALS:

Again, congratulations on a very nice paper. I hope you found the review process to be constructive and are pleased with how the manuscript was handled editorially. We look forward to future exciting submissions from your lab.

Sincerely,
